# Changes of Barley Bound Phenolics and Their Characteristics During Simulated Gastrointestinal Digestion and Colonic Fermentation *In Vitro*

**DOI:** 10.3390/foods14071114

**Published:** 2025-03-24

**Authors:** Yansheng Zhao, Fei Leng, Songtao Fan, Yiwei Ding, Tong Chen, Hongbin Zhou, Xiang Xiao

**Affiliations:** 1School of Food and Biological Engineering, Jiangsu University, Zhenjiang 212013, China; zhaoys@ujs.edu.cn (Y.Z.); leng5364118@163.com (F.L.); fansongtao@ujs.edu.cn (S.F.); dywsn44@163.com (Y.D.); 2Comprehensive Technology Centre, Zhenjiang Customs District PR China, Zhenjiang 212008, China; chentong2006@163.com (T.C.); haoranzzj@sohu.com (H.Z.)

**Keywords:** barley bound phenolics, simulated gastrointestinal digestion, colonic fermentation, gut microbiota

## Abstract

Phenolic compounds in cereals, known for their biological activities, are primarily found in a bound state within the bran. Their changes during digestion are linked to physiological activities. In this study, the dynamic changes and fermentation characteristics of barley bound phenolics (BBPs) were investigated through an *in vitro* rat gastrointestinal digestion and colonic fermentation. UPLC-HRMS revealed that the release rate of BBPs during colonic fermentation was significantly higher than that during gastric digestion (0.13%) and intestinal digestion (0.43%), reaching 5.02%. After 48 h of colonic fermentation, gallic acid and ferulic acid accounted for 35.05% and 27.84% of the total released phenolic acids, respectively. Confocal microscopy confirmed that BBPs were predominantly released in the colon. Additionally, BBPs significantly increased the content of acetate during colonic fermentation compared to the control samples, correlating with a decrease in pH value. 16S rRNA sequencing further revealed the modulatory effects of BBPs on colonic microbiota structure: BBPs significantly enhanced the Chao1 and Shannon indices of the microbiota. Notably, BBPs inhibited the growth of potentially harmful bacteria such as *Proteobacteria* and *Enterobacteriaceae* while promoting the proliferation of beneficial bacteria such as *Akkermansia* and *Bifidobacteriaceae*, thereby modulating the structure of the gut microbial community. These findings suggested that BBPs may promote gut health through prebiotic activity in the colon.

## 1. Introduction

Polyphenols are a class of secondary metabolites widely found in plants, which are of considerable attention due to their significant antioxidant and antibacterial activities [1,2,3,4,5]. These biologically active phytoconstituents exhibit a hierarchical organizational structure based on phylogenetic lineage and organ-specific biosynthesis, thereby enabling their taxonomic stratification into four principal classes: plant-derived polyphenols, fruit polyphenols, cereal polyphenols, and nut polyphenols [6]. From a structural perspective, polyphenols are further differentiated into two primary forms: free polyphenols and bound polyphenols, based on their molecular state within plant matrices [7,8]. The chemical structure and health benefits of free polyphenols have been extensively studied due to the ease of extraction and separation using organic solvents, as well as their efficient absorption and utilization by the human body [9,10,11]. In contrast, bound phenolics, which remain in the residue left after the extraction of free polyphenols from plants, are covalently linked to plant cell wall macromolecules, such as pectin, cellulose, hemicellulose, and structural proteins [12,13]. Hydrogen bonding and hydrophobic interactions, which are frequently underestimated due to the challenges associated with their separation, extraction, and resistance to degradation by digestive enzymes *in vivo*, play a significant role in the structural integrity of phenolic compounds. In recent years, substantial progress has been made in the liberation of bound phenolics from cell wall matrices through the development of diverse extraction methodologies. These include conventional chemical approaches (acid and alkaline hydrolysis), enzymatic treatments, as well as innovative techniques such as microwave-assisted extraction, ultrasound-enhanced hydrolysis, and accelerated solvent extraction [14]. It is worth noting that emerging evidence suggests that bound phenolics demonstrate superior bioactive properties compared to their free counterparts, particularly in terms of antioxidant capacity and anti-inflammatory potential. Furthermore, these compounds have been shown to exert significant therapeutic effects in mitigating colonic inflammation while simultaneously modulating gut microbiota composition through the selective promotion of beneficial bacterial populations [15].

Barley bran, a predominant by-product derived from barley processing, has long been recognized for its association with reduced risks of chronic diseases such as cardiovascular disorders, metabolic syndrome, and specific cancer types. It serves as an exceptional source of bioavailable dietary fiber with demonstrated health-promoting properties [16,17,18]. This functional component not only serves as an effective carrier of phenolic compounds to the colon but also facilitates the release of bioactive phenolics through microbial fermentation processes mediated by intestinal microbiota, thereby exerting significant physiological effects [19,20]. More notably, empirical evidence indicates that 60–85% of the phenolic content in barley bran exists in the bound form [1], which may be fundamentally associated with the health-promoting properties of bran-derived dietary fiber. Consequently, an in-depth investigation of bound phenolics in dietary fiber is scientifically important to comprehensively reveal the mechanisms by which cereal polyphenols contribute to health.

The functional efficacy of bound phenolic compounds is fundamentally governed by their bioaccessibility and subsequent bioavailability throughout the gastrointestinal (GI) tract [21]. However, bound phenolics exhibit limited bioaccessibility during digestion in the GI tract, with a substantial proportion (typically > 80%) reaching the colonic environment in association with dietary fiber, where they undergo extensive microbial biotransformation through colonic fermentation processes [22]. The gut microbiota exerts a pivotal role in the digestion, absorption, and secondary biotransformation of nutrients by modulating digestive enzyme activity, influencing intestinal epithelial barrier function, and synthesizing short-chain fatty acids [23]. This metabolic conversion is facilitated by colonic microbiota-derived enzymes, particularly those capable of cleaving ester and glycosidic linkages, thereby liberating bound phenolics from indigestible fiber matrices and potentially generating novel bioactive metabolites [24,25]. Interestingly, certain intestinal metabolites have been shown to exhibit significantly enhanced biological activity compared to their precursor compounds [26,27]. To further elucidate the functional characteristics of bound phenolics, in-depth analysis of their release, transformation, and colonic fermentation characteristics during GI digestion and colonic fermentation is necessary, which is crucial for assessing their biological significance to human health.

Based on this, the present study systematically investigated the composition of barley bran phenolics, focusing on the release of bound phenolics during *in vitro* rat gastrointestinal digestion and colonic fermentation, and further visualized the changes in microstructure and fluorescence abundance of bound phenolics during gastrointestinal digestion and colonic fermentation. Moreover, the dynamics of SCFAs production and fecal microbial community structure during colonic fermentation were analyzed in detail. These findings provide a solid scientific foundation for future development of natural and effective polyphenol functional products.

## 2. Materials and Methods

### 2.1. Materials and Reagents

Barley bran was sourced from Ruimu Biotechnology Co., Ltd., located in Yancheng, China. Standards for phenolic compounds, including caffeic acid, catechin, chlorogenic acid, syringic acid, 2,4-dihydroxybenzoic acid, salicylic acid, ferulic acid, coumaric acid, epicatechin, protocatechuic acid, and gallic acid, were obtained from Sinopharm Chemical Reagent Co., Ltd., Shanghai, China. Additionally, commercial enzymes, namely α-amylase (Termamyl SC, 120 KNU/g), protease (Alcalase 2.4 L, 2.4 AU/g), and amyloglucosidase (AMG 300 L), were procured from Novozymes (Novo Nordisk, Bagsværd, Denmark).

### 2.2. Extraction of Barley Bound Phenolics

The extraction of barley bound phenolics (BBPs) was conducted based on the procedure reported by Zhang et al. [28], with minor modifications. Initially, defatting of barley bran was achieved through extraction using hexyl hydride, yielding defatted barley bran. Subsequently, the defatted material was mixed with water at a ratio of 1:10 (w/v) and subjected to gelatinization in a water bath at 95 °C for 10 min. The mixture underwent sequential enzymatic digestion using 0.75% α-amylase (pH 5.7, 97 °C, 30 min), followed by 0.5% protease (pH 7.5, 60 °C, 60 min), and finally 0.2% amyloglucosidase (pH 4.5, 60 °C, 30 min). After enzymatic treatment, the mixture was heated at 100 °C for 15 min to inactivate enzymes and then centrifuged at 8000 rpm for 15 min (H4-21KR, Hunan Kecheng Instrument Equipment Co., Ltd., Changsha, China). The neutral precipitate was subsequently washed three times successively with 80% ethanol (1:10 w/v) followed by distilled water, and then freeze-dried to obtain BBPs (Pilot 10-15S, Beijing Boyikang Laboratory Instruments Co., Ltd., Beijing, China). The drying conditions of the freeze-drying equipment were cold trap temperature −55 °C, vacuum < 10 Pa, and plate temperature −40 °C.

### 2.3. Preparation of Hydrolyzed Bound Phenolics and Bound Phenolics-Removed Barley (RBBPs)

The preparation of hydrolyzed bound phenolics and bound phenolics-removed barley (RBBPs) was conducted using an alkaline hydrolysis procedure based on the method described in reference [29]. Specifically, the procedure entailed reacting BBPs with 800 mL of 6 M NaOH solution under an anoxic condition at ambient temperature for 4 h, with continuous stirring. Subsequently, the pH of the reaction mixture was adjusted to 2 using 2 M HCl, followed by centrifugation at 8000 rpm for 10 min. The precipitate was neutralized by washing and subsequently subjected to lyophilization to yield barley with removed bound phenolics (RBBPs). Meanwhile, the supernatant was successively extracted with ethyl acetate five times. The pooled ethyl acetate extracts were subsequently subjected to vacuum concentration and dried at 45 °C. The resulting concentrate was then dissolved in methanol to a final volume of 12 mL, yielding the hydrolyzed bound phenolics. Both the hydrolyzed bound phenolics and the RBBPs were stored at −20 °C for subsequent analytical procedures.

### 2.4. In Vitro Rat Simulated Gastrointestinal Digestion

In this study, 20 Sprague-Dawley rats (250 ± 30 g) were used to perform *in vitro* simulated gastrointestinal digestion experiments using their stomach and small intestine contents. In order to more accurately simulate the real gastrointestinal digestion environment in rats, the gastric and small intestinal contents were diluted with phosphate buffer solution (PBS). All SD rats used were purchased from the Laboratory Animal Research Center of Jiangsu University, and the experiment was completed under the supervision of the Ethics Committee for Animal Experiment at Jiangsu University (Confirmation number: UJS-IACUC-2023121602).

For the simulated gastric digestion phase, gastric contents containing sterile PBS buffer (pH 2.0, concentration 0.01 M, volume ratio of gastric contents to buffer 1/16 w/v) was used and stirred at 200 rpm for 2 h at 37 °C. Subsequently, to mimic the intestinal digestion phase, intestinal contents containing sterile PBS buffer (pH 7.0, concentration 0.01 M, volume ratio of intestinal contents to buffer 1/15 w/v) was used, and the pH was adjusted to 7.0 at 37 °C and shaken at 220 rpm for 2 h. Furthermore, blank digestion without BBPs and RBBPs was performed under the same conditions. All incubations were performed in triplicate.

### 2.5. In Vitro Colonic Fermentation

The colonic fermentation process was conducted based on the protocol established by Zhang et al. [28] with modifications. Fresh fecal samples were collected from 20 healthy Sprague-Dawley rats exhibiting no signs of gastrointestinal disorders. All fecal samples were diluted to 10% (w/v) in an anaerobic environment using aseptic PBS, followed by filtration through sterile double-layer gauze for subsequent *in vitro* fermentation experiments. The fermentation medium was sterilized via autoclaving at 121 °C for a duration of 15 min. A mixture was prepared by combining 200 mg of either BBPs or RBBPs with 5.5 mL of sterilized medium and 4.5 mL of fecal slurry in a 10 mL fermentation tube. Fermentation was conducted under anaerobic conditions at 37 °C, with a gas environment comprising 10% H_2_, 10% CO_2_, and 80% N_2_. Fermentation samples were collected at various time points (1, 6, 12, 24, and 48 h) for further analysis. The blank group contained only the medium and fecal slurry without any additional components. At each time point, samples were centrifuged at 10,000 rpm for 10 min at 4 °C (Eppendorf AG 22331 Hamburg, Eppendorf AG, Hamburg, Germany), with supernatants and precipitates stored separately at −80 °C for further analysis. Each fermentation experiment was conducted in three replicates.

### 2.6. Determination of the Composition and Content of Phenolic Compounds

The content of phenolic compounds released from undigested, digested, and fermented BBPs was determined using ultraperformance liquid chromatography coupled with high-resolution mass spectrometry (UPLC-HRMS). The analytical method was adapted from the work of Jia et al. [30] with minor modifications. The high-resolution mass spectrometer employed was the Q Exactive Plus model, manufactured by Thermo Fisher Scientific (USA), equipped with an Orbitrap mass analyzer. The chromatographic separation was achieved using a C18 column, specifically the HYPERSIL GOLD VANQUISH, with dimensions of 100 mm × 2.1 mm and a particle size of 1.9 μm. The column oven temperature was maintained at 30 °C. Mobile phase A consisted of water, while mobile phase B was methanol. The flow rate of the mobile phases was set at 0.35 mL/min, and the injection volume was 1 μL. The gradient elution program for the mobile phases was as follows: 0 min, 100% A; 0.5 min, 98% A; 5 min, 50% A; 14 min, 2% A; 16 min, 2% A; 16–19 min, 98% A. The Q-Orbitrap system of the high-resolution mass spectrometer was equipped with a heated electrospray ionization (HESI) source. Data acquisition was performed in full-scan mode, with both positive and negative ionization modes enabled. Specifically, the spray voltage was set at 3500 V for positive ion mode and 3000 V for negative ion mode. The capillary temperature was maintained at 320 °C, while the RF lens parameter was set at 50. Nitrogen was used as the collision gas. The ion source temperature was kept at 350 °C, with a sheath gas flow rate of 49 L/min and an auxiliary gas flow rate of 12 L/min. The mass range for data acquisition was set between 70 and 1050 m/z. The resolution of the mass spectrometer was 70,000 FWHM in full-scan mode and 17,500 FWHM in MS/MS mode. To ensure the accuracy and reproducibility of the chromatographic and mass spectrometric temporal and spatial information, each sample set was injected three times. The μg/g represents the weight of the various phenolic compounds released per gram of sample, where sample weight refers to the weight of the sample weighed prior to gastrointestinal digestion and colonic glycolysis.

### 2.7. Scanning Electron Microscopy (SEM)

The sample morphology of BBPs during *in vitro* gastrointestinal digestion and colonic fermentation stage were subjected to scanning electron microscopy (SEM) (S-3400N, Hitachi, Ltd., Tokyo, Japan) analysis and images were obtained at 1500× and 2500× magnification.

### 2.8. Confocal Laser Scanning Microscopy (CLSM)

The alterations in the microstructure of BBPs and the distribution of bound phenolics throughout the *in vitro* digestion and fermentation processes were examined using a confocal laser scanning microscope (CLSM) (Leica TCS SP5, Leica Camera AG, Wetzlar, Germany) equipped with a 20× objective lenses. The precipitates in different BBPs samples were dyed with a fluorescent dye solution consisting of 1.0 mg/mL Congo red. The dietary fiber image was collected with 488 nm excitation with a green fluorescence and the phenolics were collected with 408 nm excitation that exhibited a blue fluorescence.

### 2.9. Determination of pH and SCFAs

The pH levels of the fermentation samples were determined via a pH meter (Shanghai Leici Apparatus Corp., Shanghai, China). The concentrations of SCFAs were determined based on a previously described methodology, incorporating several modifications as outlined in reference [31]. Specifically, the fermentation broth was centrifuged at 12,000× *g* rpm for 5 min. Then 500 μL of the supernatant was aspirated and 35 μL of 45% phosphoric acid solution was added. The levels of SCFAs were determined via gas chromatography (ShimadzuGC-2010Plus, Tokyo, Japan) using a DB-FFAP column (30 m × 0.25 mm × 0.25 μm) and the carrier gas was high-purity nitrogen at a flow rate of 1 mL/min. The gas was high purity hydrogen at a flow rate of 40 mL/min, and the auxiliary gas was air at a flow rate of 300 mL/min. The injection volume was 0.6 μL, the injection port temperature was 250 °C, the injection mode was split injection, the split ratio was 20:1, the septum purge was 3 mL/min, and the detector temperature was 220 °C.

### 2.10. Gut Microbiota Analysis

The gut microbiota analysis was carried out by Shanghai Biotree Biomedical Technology Co., Ltd. (Shanghai, China). For the procedure, the fermentation broth gathered at the 48-h mark was utilized. Total fecal microbial DNA was extracted using the Fecal Genome DNA Extraction Kit (AU46111-96, BioTeke, Beijing, China) following the manufacturer’s protocol. Subsequent steps included PCR amplification, product purification, library preparation and quality control, and sequencing. The sequencing was performed on the Illumina NovaSeq platform, and data analysis followed. The PCR amplification used the universal primer pair 341F/805R (341F: 5′-CCTACGGGNGGCWGCAG-3′; 805R: 5′-GACTACHVGGGTATCTAATCC-3′) to target the V3-V4 region of the 16S rRNA gene.

### 2.11. Statistical Analysis

All experiments were conducted in triplicate unless otherwise noted, with results presented as the mean ± standard deviation (SD). Graphical illustrations were created using GraphPad Prism 9.5. Statistical analysis was performed via one-way analysis of variance (ANOVA) followed by Duncan’s multiple range test, utilizing SPSS 22.0 (SPSS Inc., Chicago, IL, USA). Furthermore, α-diversity was evaluated using the Wilcoxon test. A *p*-value below 0.05 was deemed to indicate statistical significance.

## 3. Results and Discussions

### 3.1. Quantitative Analysis of Digested and Fermented BBPs

The composition and quantitative profiles of bound phenolic compounds in undigested BBPs, along with those released during sequential *in vitro* rat gastrointestinal digestion and colonic fermentation, were summarized and analyzed in Table 1 and Table 2. In this study, nine phenolic acids and two flavonoid compounds, including catechins and epicatechin, in BBPs were quantitatively analyzed. As shown in Table 1, gallic acid and syringic acid were the main phenolic acids in the undigested BBPs, which accounted for 32.71% and 24.63% of the total phenolic content, respectively. In addition, gallic acid and syringic acid were also the main phenolic acids liberated after *in vitro* gastric digestion. Notably, ferulic acid is released in the highest amount during gastric digestion. After *in vitro* intestinal digestion, phenolic acid release was significantly increased compared to the gastric digestion phase, especially gallic acid, epicatechin, and ferulic acid. However, after *in vitro* simulated digestion, only 60.32 ± 0.02 μg/g DW of polyphenols were released at the end of the intestinal phase, accounting for 0.56% of the total phenolics in the undigested BBPs; which suggested that the majority of the bound phenolics in BBPs are not hydrolyzed by the digestive enzymes. Similarly, it has been shown that only 1.18% of polyphenols were released following gastrointestinal digestion, whereas more than 95% of polyphenols could be released by enzymes secreted by colonic microflora [32]. Swallah et al. conducted an in-depth analysis and demonstrated that the colonic microbiome has the metabolic capacity to further degrade these polyphenol structures into smaller, more readily absorbable molecules, a process of crucial significance as it plays a pivotal role in promoting human health [33].

Data from further investigation into the composition of released phenolics during colonic fermentation stage are presented in Table 2. Gallic acid was the main phenolic compound released from BBPs during the colonic fermentation and its content peaked at the 48 h of fermentation point. Additionally, ferulic acid and epicatechin were positively correlated with the total phenolic content and increased significantly with fermentation time. This suggested that bound phenolics can be released better during fermentation stage by microorganisms. It was found that the amounts of compounds such as caffeic acid, catechin, chlorogenic acid, and syringic acid decreased during the colonic fermentation phase, which indicated that they may be metabolized to other small molecules or rapidly degraded by the gut microbiota. For example, syringic acid can undergo demethylation and dehydroxylation to form substances such as gallic acid, protocatechuic acid, or p-hydroxybenzoic acid under the action of colonic microbiota and related esterases [34]. Interestingly, cumaric acid, which was less released during *in vitro* simulated digestion, was found in high concentration in colonic fermentations. This may be due to microbial fermentation and transformation. Colonic fermentation of fermented sorghum using rat fecal microbiota demonstrated that the reduction in phenolic compound content during fermentation is likely associated with the metabolism of these compounds by the gut microbiota, leading to the formation of new compounds, including novel phenolic derivatives and non-aromatic products [35]. The 2, 4-dihydroxybenzoic acid levels in the 48 h fermentation supernatants were higher than those in the 24 h fermentation, and fermentation duration has been demonstrated to exert a significant influence on the quantity of phenolic acids released. The terminal microbial metabolic byproducts, encompassing phenolic compounds, may consist of either straightforward phenolic entities or non-phenolic constituents, exemplified by benzoic acid and phenylpropionic acid. During the colonic fermentation process, rice bran dietary fiber-bound phenols were detected to contain 10 phenolic compounds, and the potential catabolic pathways of these phenolic compounds were further analyzed. For example, ferulic acid can be metabolized into caffeic acid through methylation, while caffeic acid may undergo de-esterification and rapid dihydroxylation reactions, ultimately resulting in the formation of hydroxyphenylpropionic acid. Additionally, hydroxybenzoic acid may also be derived from the conversion of vanillic acid [28]. The findings of this study indicate that the bound phenolics originating from BBPs experience dynamic release, transformation, and utilization by the gut microbiota. It is worth noting that the present investigation was restricted to the examination of changes in the primary phenolic acids during the *in vitro* processes of gastrointestinal digestion and colonic fermentation. To achieve a more comprehensive comprehension, further research is essential to elucidate the detailed profile of phenolic metabolites produced.

### 3.2. The Microstructural Changes in BBPs During Different Digestion and Colonic Fermentation Stages

The microstructures of BBPs that were undigested and after *in vitro* digestion and colonic fermentation were analyzed using SEM, which is an effective way to visualize the microstructural changes in dietary fiber. As shown in Figure 1, the microstructure of undigested BBPs exhibited integrity and smoothness, while after *in vitro* digestion, the sheet-like structure of BBPs displayed slight curling, but remained largely intact, showing a slightly loose block-like structure and flake-like structure in some places. This phenomenon may be attributed to the enzymatic degradation of a minor fraction of dietary fiber occurring within the gastric and intestinal phases of digestion [36]. However, after colonic fermentation, the integrity of the overall structure of dietary fiber was disrupted. The smooth, layered structure of BBPs underwent transformation into a highly porous, lamellar configuration, which significantly increased its specific surface area relative to its pre-fermentation state in the colon. This structural alteration likely facilitates greater exposure of BBPs to carbohydrate hydrolases secreted by the gut microbiota. Consistent with prior research [37], the gut microbiota employs diverse microbial enzymes to catalyze the degradation of these complex molecules, thereby creating more accessible sites for the release of phenolic compounds [32].

### 3.3. The Changes in the Fluorescence Abundance of Bound Phenolics During Different Digestion and Colonic Fermentation Stages

CLSM was used to visualize the changes in the fluorescence abundance of phenolics of BBPs during the *in vitro* digestion and fermentation process. The blue signal on the BBPs’ surface was caused by the spontaneous fluorescence of phenolic compounds and the green signal on the BBPs’ surface was caused by the spontaneous fluorescence of dietary fibers. As shown in Figure 2(A-1), the fluorescence abundance of phenolic compounds was clearly visible in the undigested stage. However, after gastrointestinal digestion, its fluorescence abundance was slightly weakened (Figure 2(B-1,C-1)), indicating that during the gastrointestinal digestion process, a small portion of bound phenolics was degraded by gastrointestinal digestive enzymes, while the majority of bound phenolics were degraded during the colonic fermentation stage, during which the blue fluorescence was not visible (Figure 2(D-1)). The colon contains a large number of intestinal flora, whose released esterases and other substances can effectively degrade bound phenolics [37]. Furthermore, the disruption of the rigid structure of the dietary fibers in barley bran can make it easier for microbial enzymes to release more of the bound phenolics [38].

### 3.4. Effects of BBPs and RBBPs on SCFAs During Fermentation

Short-chain fatty acids (SCFAs) are the predominant metabolic products derived from the fermentation activities of the gut microbiota, encompassing acetate, propionate, butyrate, valerate, and isovalerate. Among these, acetate, propionate, and butyrate constitute the major SCFA fractions, accounting for more than 90% of the total SCFA pool [39]. These metabolites serve as an energy source for the colonic mucosa and play a crucial role in maintaining the integrity of intestinal epithelial cells [40]. As illustrated in Figure 3A–D, this study measured the concentrations of acetate, propionate, and butyrate at various fermentation time points to evaluate the fermentation characteristics of BBPs. The results indicated that from 1 to 24 h, the total concentration of SCFAs exhibited a varying degree of increase with the extension of fermentation time. Notably, the total SCFA concentration in the BBPs group was significantly higher than that in the RBBPs group and the blank control group, suggesting that the bound phenolic compounds in BBPs can promote the generation of more SCFAs during colonic fermentation. This finding further underscores the significant role of bound phenolic compounds in gut microbial metabolism, particularly in facilitating the production of beneficial metabolites. During the colonic fermentation process, triticale insoluble dietary fiber containing bound phenolic compounds exhibited a higher yield of SCFAs compared to dephenolized dietary fiber. This finding suggests that the bound phenolic compounds can interact synergistically with the dietary fiber matrix, thereby significantly enhancing the production of SCFAs in the colon [22]. In addition, compared with free catechins and dephenolic indigestible dextrin fiber, the combined polyphenols of highland barley fiber more effectively enhanced the total content of SCFAs [41]. Notably, total SCFA levels peaked at 24 h and then decreased slightly at 48 h. The underlying mechanisms for this phenomenon may include nutrient depletion mediated by bacterial colonization, with SCFA produced during the initial stage of colonic fermentation being subsequently metabolized. Additionally, alterations in the composition of the intestinal microbiota throughout the fermentation process may exert deleterious effects on SCFA [42]. The comparable patterns were discerned during the *in vitro* fermentation of resistant starch [43]. Individually, compared with propionic acid, acetic acid and butyric acid mainly contributed to the difference in total SCFAs. After 48 h of fermentation, there was a significant difference in acetic acid and butyric acid content between the BBPs and RBBPs groups (*p* < 0.05). The concentration of acetic acid and butyric acid in BBPs increased to 19.95 mM and 2.674 mM, values which were 26.7% and 36.9% higher than those in the RBBPs group after 48 h of fermentation (*p* < 0.05), respectively. In addition, the propionic acid value (1.761 mM) was significantly higher than that in the blank and RBBPs groups (*p* < 0.01). These results suggest that the gut microbiota could influence the biological characteristics of phenolic acids through fermenting bound phenolics in the plant matrix to produce SCFAs.

### 3.5. Effects of BBPs and RBBPs on pH During Fermentation

During the colonic fermentation phase, the pH value serves as one of the key indicators of fermentation progress. As shown in Figure 3E, the pH value of the BBPs group was significantly higher than that of the blank and RBBPs groups after 1 h of fermentation, which may be attributed to the gradual release of bound phenolic compounds facilitated by microbial activity. After 12 h of fermentation, the pH value of the BBPs group was significantly lower than that of the blank and RBBPs groups (*p* < 0.0001), ranging from 4.43 to 5.46. This change is likely due to the release of bound phenolic compounds and the production of SCFAs, which contribute to the decrease in pH value [44]. Notably, after 48 h of fermentation, the pH values of all three groups exhibited a slight increase to varying degrees, consistent with the trend of total SCFA concentration. This suggests that the accumulation of SCFAs is the primary driver of pH changes [45].

### 3.6. Effects of BBPs and RBBPs Fermentation on Gut Microbiota

The possible effects of bound phenolics released from BBPs on the gut microbiota were further analyzed by executing 16S rRNA sequencing analysis to investigate changes in the fecal microbiota communities of BBPs and RBBPs slurry. α-diversity and OTU indices can be used to assess the richness and diversity of communities. As shown in Figure 4A, alpha diversity indices were measured, including the Simpson index, Chao1 index, observed species, and the Shannon index. At 48 h of fermentation, the BBPs group exhibited the highest α-diversity compared to the blank and RBBPs groups, indicating that BBPs can significantly enhance the richness and diversity of the gut microbiota. Meanwhile, Figure 4B showed that the number of OTU numbers shared by BBPs and blank groups was higher than that of the RBBPs group, indicating that the BBPs group had a higher species richness. In addition, the bound phenolics significantly affected the microflora structures (Figure 4C). The PCOA plots revealed distinct clustering patterns among the three groups, demonstrating a statistically significant separation between the different fermentation groups. The results indicated that the increase in the richness and diversity of the gut microbiome is closely related to the release of bound phenolics from BBPs.

To elucidate the potential influence of bound phenolics released from BBPs on microbial composition, we examined the characteristics of the bacterial community across various sample groups at multiple taxonomic levels. As illustrated in Figure 5A, at the phylum level, consistent with other studies, *Firmicutes*, *Bacteroidetes*, *Proteobacteria*, and *Actinobacteria* collectively account for approximately 99% of the total bacteria in the gut, constituting the majority of the human colonic flora. Compared to the blank group, both the BBPs and RBBPs groups significantly reduced the relative abundance of the *Proteobacteria* phylum. Notably, the BBPs group exhibited a significant increase in the relative abundance of the *Actinobacteriota* and *Bacteroidota* phyla compared to the RBBPs group. Despite their minor proportions, *Actinobacteriota*, one of the four major phyla of the gut microbiota, plays a crucial role in maintaining intestinal homeostasis [46]. The *Bacteroidetes* phylum, known for its proficiency in polyphenol uptake and metabolism, likely explains the significant role of polyphenols in this context. The continuous increase in the relative abundance of the *Bacteroidota* phylum in the BBPs group may be associated with the continuous release of ferulic acid from BBPs during *in vitro* fecal fermentation by *Bacteroidetes* colonies, which secrete feruloyl esterase and exocellulase. Xie et al. demonstrated that polyphenols released from mung bean hull fermentation can promote the growth of *Bacteroidetes*, thereby favorably enhancing intestinal homeostasis [47]. Additionally, previous studies confirmed that *Bacteroidetes* can secrete vesicle hydrolases involved in numerous important metabolic activities in the human colon, such as degrading dietary fiber, starch, and protein to provide nutrients, secreting SCFAs, and maintaining the stability of the intestinal environment through metabolic products [48]. These findings indicate that the modulation of microbial community balance by bound phenolics and their metabolites during colonic fermentation is a key distinguishing feature of the BBPs group compared to the RBBPs group.

At the genus level, both the BBPs and RBBPs groups exhibited increased relative abundances of *Lactobacillus* and decreased the relative abundances of *Escherichia-Shigella* compared to the blank group. Research has shown that *Lactobacillus* possesses multiple functions within the human body, including maintaining intestinal flora balance, enhancing immunity, and promoting the absorption of nutrients [49]. An increased relative abundance of *Lactobacillus* may be related to the degradation of dietary fiber and the release and transformation of bound phenolics during colonic fermentation [50]. The *Escherichia-Shigella* genus is a major pathogen responsible for human intestinal diseases, capable of altering the connections between intestinal epithelial cells, damaging the adhesion between intestinal epithelial cells and the intestinal mucosa, and altering the structure of the intestinal mucosa [51]. These destructive effects increase intestinal barrier permeability, allowing bacteria, toxins, and other harmful substances to pass through the intestinal barrier into the circulatory system, triggering inflammatory responses and other intestinal-related diseases. The significant inhibition of their growth may be due to the selective suppression of harmful bacteria by the dietary fiber and bound phenols released from barley bran. More importantly, compared to the RBBPs group, the BBPs group showed a significant increase in the relative abundance of the *Muribaculaceae*, a bacterial family within the *Bacteroidales* order, which can produce short-chain fatty acids through both endogenous (mucin glycans) and exogenous polysaccharides (dietary fiber) [52]. Research has indicated that an increased abundance of *Muribaculaceae*, a potential probiotic family, is associated with the alleviation of inflammatory bowel disease, obesity, and type 2 diabetes [53]. Furthermore, as shown in Figure 5B,C, the BBPs group exhibited a significant increase in the relative abundance of the *Bifidobacterium* genus, indicating that bound phenolics and their metabolites have a significantly promoting effect on *Bifidobacterium*. *Bifidobacteria* are beneficial bacteria that adhere to the intestinal mucosal surface, with functions including regulating the intestinal flora, alleviating colonic inflammation, and regulating glucose and lipid metabolism [54]. Research by Zahid et al. found that mango peel powder after colonic fermentation significantly increased the number of *Bifidobacterium*, likely due to the ability of intestinal microbes to convert complex phenolic substances into low molecular weight components, which exert their beneficial effects through their probiotic-like actions, regulating beneficial intestinal flora such as *Lactobacillus* and *Bifidobacterium* [42]. Therefore, the release and metabolism of bound phenolics have the potential to modulate the equilibrium of the colonic flora. This regulation may be achieved by promoting the growth of beneficial bacteria and inhibiting pathogenic bacteria, thus being able to regulate the delicate balance of the gut microbial ecosystem.

### 3.7. Correlation Between Phenolics SCFAs and Gut Microbiota

Figure 6 illustrates the correlations among the production of major phenolic compounds, SCFAs, and gut microbiota in the blank, BBPs, and RBBPs groups using Spearman correlation analysis. According to the heatmap, *Bifidobacterium* and *Adlercreutzia* were significantly positively correlated with the release of benzoic acid, gallic acid, epicatechin, and ferulic acid, and a significantly negative relationship with *Desulfovibrio* and *Escherichia-shigella*. Combining with Figure 5, we saw that BBPs could regulate the composition of the human microbiota through its phenolics; thus, they may play a key role in regulating intestinal health. In addition, the release of bound phenolic compounds by BBPs during colonic fermentation, mediated via dietary fiber as a carrier, may account for the differences in gut bacterial composition observed between the BBPs and RBBPs groups.

In terms of production of SCFAs, *Ruminococcus* and *Alistipes* were significantly positively correlated with the production of total SCFAs. *Muribaculaceae* was significantly positively correlated with the production of butyrate and propanoic acid. *Alistipes* exhibited a strong positive association with the production of acetic acid and propanoic acid. Combining with Figure 3, we saw that BBPs increased the production of SCFAs by promoting the growth of *Ruminococcus*, *Muribaculaceae,* and *Alistipes*.

## 4. Conclusions

In summary, during the simulated *in vitro* rat gastrointestinal digestion and colonic fermentation processes, only trace amounts of phenolic compounds were degraded at the gastrointestinal digestion stage. The microbial-mediated fecal fermentation served as the predominant pathway for the release and transformation of barley bound phenolics. Confocal laser scanning microscopy experiments further confirmed that the colon was the site for the release of barley bound phenolics. During the fermentation process, the release of gallic acid and ferulic acid was significant, and the distribution of phenolic compounds experiencing release exhibited dynamic changes at different fermentation time points. The dynamic changes of phenolics may be the result of their release by colonic microbiota or their transformation. The intestinal microbiota disrupted the interaction between the bound phenolics and the carrier dietary fiber, and the released polyphenols and their catabolic products promoted the synthesis of SCFAs. Additionally, the released bound phenolic substances enhanced the relative abundance of *Bifidobacteria* and *Lactobacilli*, while decreasing that of *Escherichia-Shigella*. The results of this study indicated that the bound phenolic compounds in barley bran may be the primary constituents contributing to its beneficial health effects. To definitively establish the prebiotic efficacy of BBPs, additional comprehensive investigations involving human subjects, animal models, or cellular experiments are warranted.

## Figures and Tables

**Figure 1 foods-14-01114-f001:**
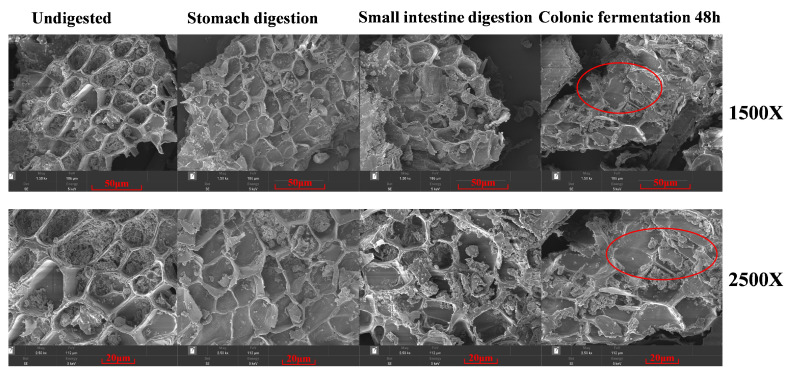
SEM micrographs of undigested, digested, and fecal fermented BBP samples (The red circles indicate that the integrity of the overall structure of dietary fiber was disrupted).

**Figure 2 foods-14-01114-f002:**
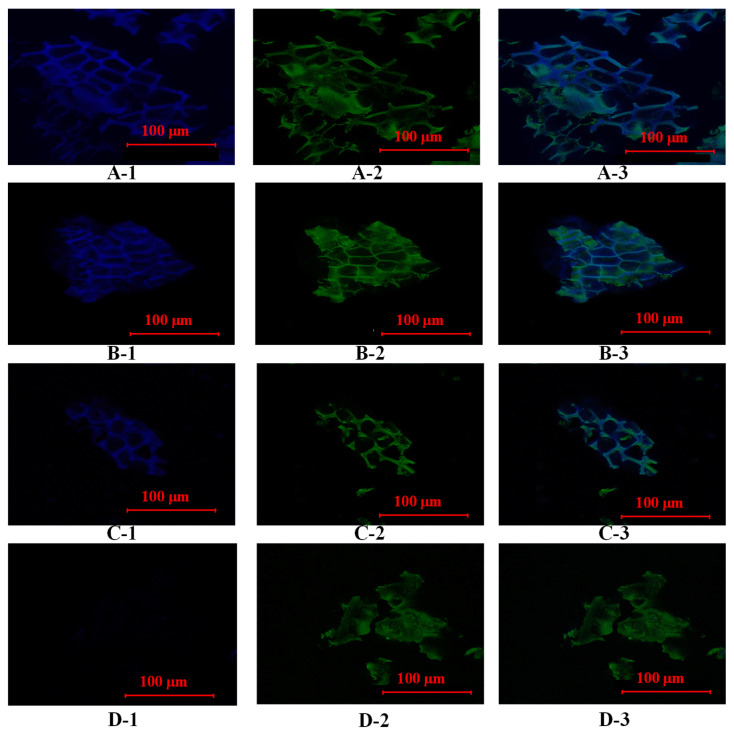
Confocal micrographs of undigested, digested, and fermented BBPs samples. (**A-1**) Fluorescence image of the bound phenolics of Undigested BBPs; (**A-2**) Fluorescence image of the dietary fibers of Undigested BBPs; (**A-3**) Fused image of fluorescence images of bound phenolics and dietary fibers of Undigested BBPs; (**B-1**) Fluorescence image of the bound phenolics of BBPs after *in vitro* stomach digestion; (**B-2**) Fluorescence image of the dietary fibers of BBPs after *in vitro* stomach digestion; (**B-3**) Fused image of fluorescence images of bound phenolics and dietary fibers of BBPs after *in vitro* stomach digestion; (**C-1**) Fluorescence image of the bound phenolics of BBPs after *in vitro* intestine digestion; (**C-2**) Fluorescence image of the dietary fibers of BBPs after *in vitro* intestine digestion; (**C-3**) Fused image of fluorescence images of bound phenolics and dietary fibers of BBPs after *in vitro* intestine digestion; (**D-1**) Fluorescence image of the bound phenolics of BBPs after colonic fermentation 48 h; (**D-2**) Fluorescence image of the dietary fibers of BBPs after colonic fermentation 48 h; (**D-3**) Fused image of fluorescence images of bound phenolics and dietary fibers of BBPs after colonic fermentation 48 h. Dietary fiber was dyed in green and phenolics were dyed in blue due to their autofluorescence.

**Figure 3 foods-14-01114-f003:**
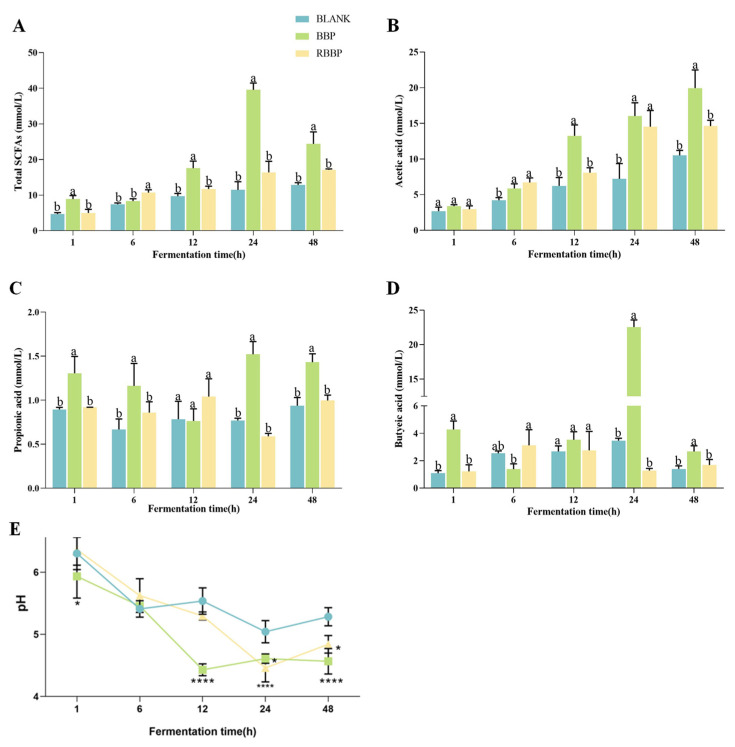
Changes in SCFAs and pH during colonic fermentation of barley bound phenolics. (**A**) Content of total SCFAs in fermentation solution. (**B**) Content of acetic acid in fermentation solution. (**C**) Content of propionic acid in fermentation solution. (**D**) Content of butyric acid in the fermentation solution. (**E**) Changes in pH during in the fermentation solution. * Indicates comparison with blank group, *: *p* < 0.05, ****: *p* < 0.0001. Different letters represent significant differences between the two groups (*p* < 0.05).

**Figure 4 foods-14-01114-f004:**
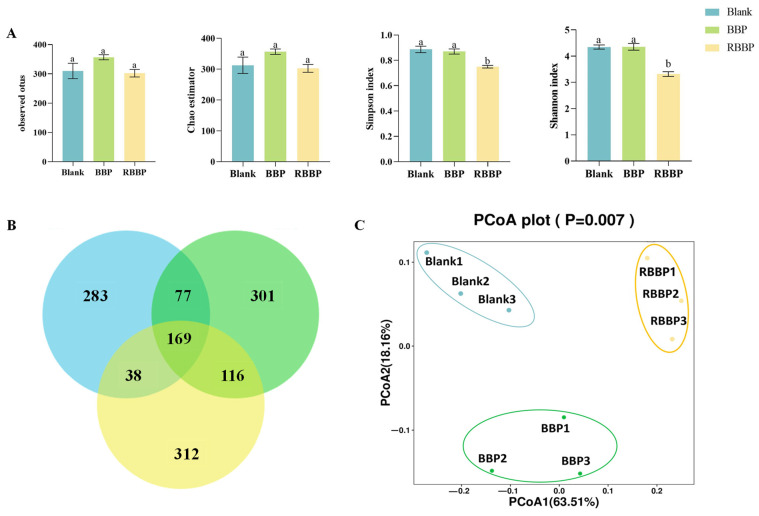
Fermentation behavior of BBPs and RBBPs in gut microbiota diversity. (**A**) Alpha diversity analysis of gut microbiota: observed OTU index curve, Chao1 index curve, Simpson index curve, Shannon index curve; (**B**) OTU diagram; (**C**) Bray–Curtis PCoA analysis for all samples in terms of relative abundance Different letters represent significant differences between the two groups (*p* < 0.05).

**Figure 5 foods-14-01114-f005:**
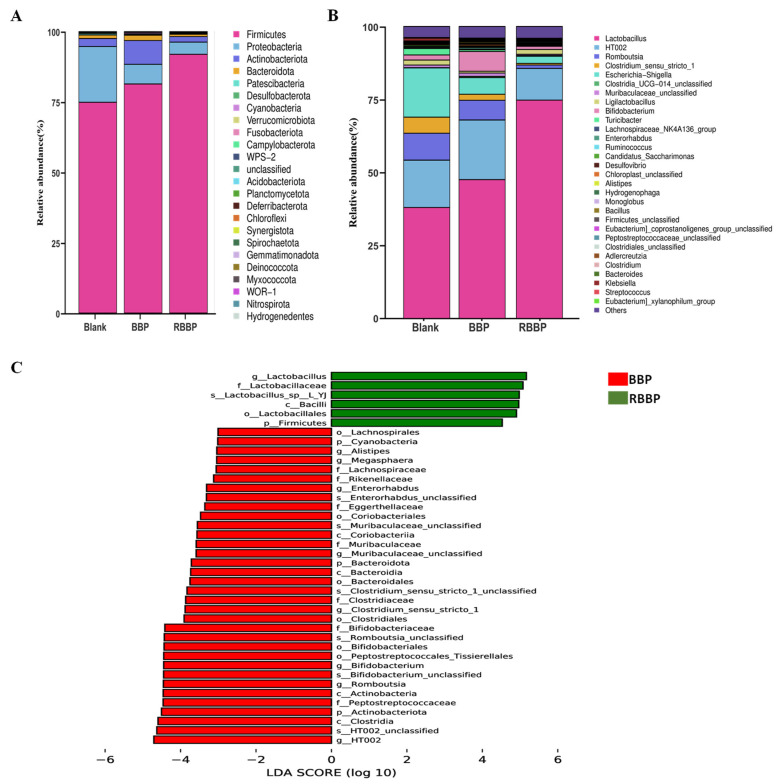
The gut microbiota composition of fecal fermented BBPs and RBBPs at different levels. (**A**) Changes in gut microbiota at the phylum level. (**B**) Changes in gut microbiota at the genus level. (**C**) Circos graphs showing the distribution proportion of dominant gut microbiota in each group on the genus level.

**Figure 6 foods-14-01114-f006:**
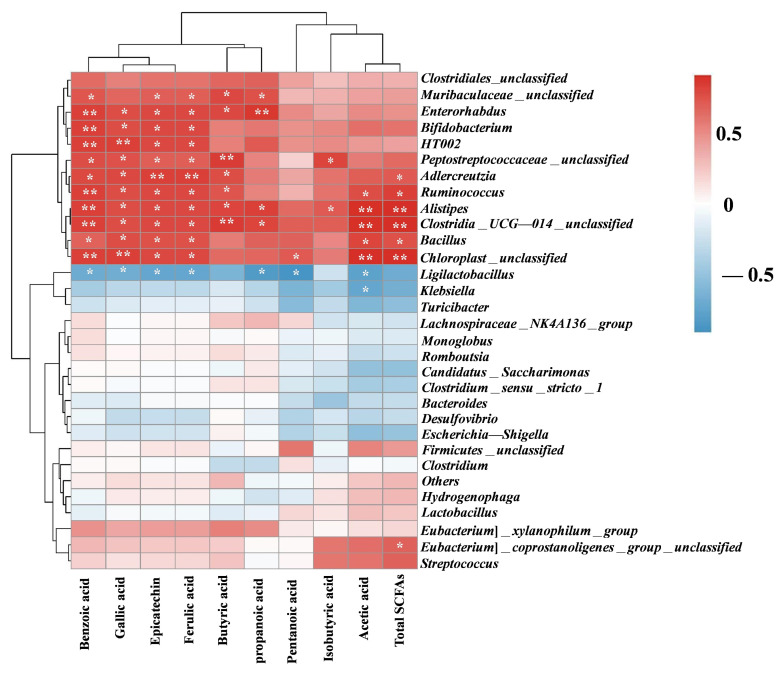
Spearman correlation analysis of the top 30 genera and metabolites in relative abundance. Red color represents a positive correlation, and blue color represents a negative correlation. The shades of color represent the level of significance. * *p* < 0.05; ** *p* < 0.01.

**Table 1 foods-14-01114-t001:** Quantitative results of bound phenolics released from BBPs during *in vitro* digestion stage (μg/g).

Compounds	Undigested BBPs	Gastric Digestion	Intestinal Digestion
Caffeic acid	372.00 ± 12.25 ^a^	0.32 ± 0.01 ^b^	0.41 ± 0.01 ^b^
Catechin	195.00 ± 7.35 ^a^	1.22 ± 0.01 ^b^	1.99 ± 0.01 ^b^
Chlorogenic acid	657.00 ± 22.05 ^a^	0.82 ± 0.02 ^b^	1.52 ± 0.02 ^b^
Syringic acid	2654.00 ± 89.41 ^a^	1.75 ± 0.02 ^b^	3.28 ± 0.07 ^b^
2,4-dihydroxybenzoic acid	582.00 ± 19.6 ^a^	0.26 ± 0.01 ^b^	0.54 ± 0.01 ^b^
Salicylic acid	945.00 ± 31.84 ^a^	0.47 ± 0.01 ^b^	1.09 ± 0.02 ^b^
Ferulic acid	270.00 ± 9.8 ^a^	3.23 ± 0.01 ^c^	11.93 ± 0.16 ^b^
Coumaric acid	999.00 ± 34.29 ^a^	1.04 ± 0.01 ^c^	4.33 ± 0.05 ^b^
Epicatechin	384.00 ± 12.25 ^a^	1.79 ± 0.01 ^c^	8.19 ± 0.11 ^c^
Protocatechuic acid	192.00 ± 7.35 ^a^	0.10 ± 0.01 ^c^	0.48 ± 0.01 ^b^
Gallic acid	3524.00 ± 118.8 ^a^	2.61 ± 0.00 ^c^	13.01 ± 0.3 ^b^

BBPs: barley bound phenolics. Values with different letters in the same line denote a significant difference (*p* < 0.05).

**Table 2 foods-14-01114-t002:** Quantitative results of bound phenolics released from BBPs during colonic fermentation stage (μg/g).

Compounds	Colonic Fermentation				
	1 h	6 h	12 h	24 h	48 h
Caffeic acid	0.58 ± 0.03 ^a^	0.30 ± 0.00 ^b^	0.29 ± 0.01 ^b^	0.22 ± 0.00 ^c^	0.13 ± 0.00 ^d^
Catechin	4.38 ± 0.09 ^a^	4.01 ± 0.12 ^b^	2.55 ± 0.04 ^c^	1.65 ± 0.03 ^d^	0.99 ± 0.03 ^e^
Chlorogenic acid	3.23 ± 0.04 ^a^	2.73 ± 0.05 ^b^	2.30 ± 0.05 ^c^	1.93 ± 0.06 ^d^	1.59 ± 0.05 ^e^
Syringic acid	7.95 ± 0.18 ^a^	3.43 ± 0.06 ^b^	2.83 ± 0.12 ^c^	1.70 ± 0.05 ^d^	0.90 ± 0.01 ^e^
2,4-dihydroxybenzoic acid	1.51 ± 0.02 ^e^	2.64 ± 0.03 ^d^	3.98 ± 0.09 ^c^	5.48 ± 0.12 ^b^	10.36 ± 0.20 ^a^
Salicylic acid	3.49 ± 0.05 ^e^	7.41 ± 0.05 ^d^	7.94 ± 0.26 ^c^	9.98 ± 0.09 ^b^	12.65 ± 0.24 ^a^
Ferulic acid	38.58 ± 0.53 ^e^	54.65 ± 0.28 ^d^	74.64 ± 0.53 ^c^	98.13 ± 0.65 ^b^	154.06 ± 3.60 ^a^
Coumaric acid	12.55 ± 0.06 ^e^	15.44 ± 0.21 ^d^	19.2 ± 0.48 ^c^	27.93 ± 0.50 ^b^	47.3 ± 0.85 ^a^
Epicatechin	23.78 ± 0.67 ^d^	50.32 ± 1.73 ^c^	52.87 ± 0.69 ^c^	63.48 ± 0.75 ^b^	124.54 ± 3.85 ^a^
Protocatechuic acid	1.67 ± 0.04 ^d^	2.44 ± 0.07 ^c^	2.51 ± 0.10 ^c^	3.23 ± 0.00 ^b^	6.93 ± 0.16 ^a^
Gallic acid	45.32 ± 1.49 ^e^	72.46 ± 0.20 ^d^	88.51 ± 1.08 ^c^	110.03 ± 1.06 ^b^	193.98 ± 2.21 ^a^

BBPs: barley bound phenolics. Values with different letters in the same line denote a significant difference (*p* < 0.05).

## Data Availability

The original contributions presented in the study are included in the article, further inquiries can be directed to the corresponding author.

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
