# Peer review of "Changes of Barley Bound Phenolics and Their Characteristics During Simulated Gastrointestinal Digestion and Colonic Fermentation In Vitro"

_foods, 2025, doi:10.3390/foods14071114_

Round 1

Reviewer 1 Report

Comments and Suggestions for Authors

Comments to the Authors

General remarks:

The manuscript foods- 3514792 investigate and address the influence of gastrointestinal digestion and colonic fermentation in vitro on changes of barley bound phenolics and their characteristics. The authors conducted a lot of analyses relevant to the research and helpful in complete understanding of the changes in barley bound phenolic compounds bringing novelty to the topic. The research topic is relevant for further understanding of the role of bound phenolic in raw materials as well as the food products produced.

Below you will find detailed comments.

Abstract: Is of good quality.

Introduction: Is of good quality.

Materials and methods are relevant and appropriate for the research.

Page 3, line 123 – please include model of centrifuge used, manufacturer, city and country.

Page 3, line 125 – please include model of lyophilisator, manufacturer, city and country of production as well as drying conditions used.

Page 3, line 131 - please include model of centrifuge used, manufacturer, city and country.

Page 3, line 132 – please include model of lyophilisator, manufacturer, city and country of production as well as drying conditions used.

Page 4, line 165 - please include model of centrifuge used, manufacturer, city and country.

Page 4, line 187 – Please include model of electron microscope used, manufacturer, city and country.

Page 4, line 191 – Please include model of confocal laser scanning microscope used, manufacturer, city and contra.

Results & Discussion: well written and supported by existing literature.

Page 6, lines 264-268 – Are there similar results on other cereals or just kiwi fruit? Please complement.

Page 6, lines 268-271 – Any similar observation within existing literature? Please complement.

Page 9, lines 338-340 – Are there similar results on other cereals or just Rosa roxburghii fruit? Please complement.

Conclusion: Is of good quality and consistent with the results presented.

Figures and Tables are well organised and presented.

References are relevant to the topic and up to date.

Author Response

Comments 1: Page 3, line 123 – please include model of centrifuge used, manufacturer, city and country.

Response 1: Thank you for your kind comments. The model of the centrifuge is H4-21KR, and the manufacturer is Hunan Kecheng Instrument Equipment Co., Ltd., Changsha, Hunan Province, and we have supplemented this information in the manuscript. (Line 116-117)

Comments 2: Page 3, line 125 – please include model of lyophilisator, manufacturer, city and country of production as well as drying conditions used.

Response 2: Thank you for your kind comments. The model of the freeze-drying equipment is Pilot 10-15S, and the manufacturer is Beijing Boyikang Laboratory Instruments co., Ltd, Beijing. The drying conditions of the freeze-drying equipment were cold trap temperature: -55 ℃, vacuum: <10 Pa, and plate temperature: -40 ℃, and we have supplemented this information in the manuscript. (Line 119-122)

Comments 3: Page 3, line 131 - please include model of centrifuge used, manufacturer, city and country.

Response 3: Thank you for your kind comments. The centrifuge details are the same as those used in Line 116-117.

Comments 4: Page 3, line 132 – please include model of lyophilisator, manufacturer, city and country of production as well as drying conditions used.

Response 4: Thank you for your kind comments. The freeze-drying equipment details are the same as those used in Line 119-122.

Comments 5: Page 4, line 165 - please include model of centrifuge used, manufacturer, city and country.

Response 5: Thank you for your kind comments. The model of the centrifuge is Eppendorf AG 22331 Hamburg, and the manufacturer is Eppendorf AG, Hamburg, Germany, and we have supplemented this information in the manuscript. (Line 166-167)

Comments 6: Page 4, line 187 – Please include model of electron microscope used, manufacturer, city and country.

Response 6: Thank you for your kind comments. The model of the SEM is S-3400N, and the manufacturer is Hitachi, Ltd., Tokyo, Japan, and we have supplemented this information in the manuscript. (Line 199-200)

Comments 7: Page 4, line 191 – Please include model of confocal laser scanning microscope used, manufacturer, city and country.

Response 7: Thank you for your kind comments. The model of the confocal laser scanning microscope is Leica TCS SP5, and the manufacturer is Leica Camera AG, Wetzlar, Germany, and we have supplemented this information in the manuscript. (Line 205-206)

Comments 8: Page 6, lines 264-268 – Are there similar results on other cereals or just kiwi fruit? Please complement.

Response 8: Thank you for your kind suggestions. We have added similar results related to other cereals. (Line 280-284)

Comments 9: Page 6, lines 268-271 – Any similar observation within existing literature? Please complement.

Response 9: Thank you for your kind suggestions. We have added similar observation related to other cereals. (Line 289-296)

Comments 10: Page 9, lines 338-340 – Are there similar results on other cereals or just Rosa roxburghii fruit? Please complement.

Response 10: Thank you for your kind suggestions. We have added similar results related to other cereals. (Line 359-366)

Reviewer 2 Report

Comments and Suggestions for Authors

The manuscript is comprehensive and well-structured.

It is hard to have a proper separation of Catechin and epicatechin from phenolic acids in single run. Peaks may be overlapping.. Can you explain how is the chromatogram looks and what is the retention times.

How is the calculation for the phenolics expressed as ug/g for the gastric and intestinal digestion as these media is liquid.. is it referred back to the original grain weight?

L 376  what is OTU indices, please defined..

L382 number of OUT … Is this a typo error ! it should be OUT !  also Check figure 4 as well it is written as OUT.

The manuscript has 6 figures, it may be a good idea to have a supplementary material.

Reference format need to be revised.

Author Response

Comments 1: It is hard to have a proper separation of Catechin and epicatechin from phenolic acids in single run. Peaks may be overlapping. Can you explain how is the chromatogram looks and what is the retention times.

Response 1: Thank you for your kind comments. In this study, ultra-performance liquid chromatography-high resolution mass spectrometry (UPLC-HRMS) technology was used to qualitatively and quantitatively analyze phenolic substances with the help of high-resolution mass spectrometry database and a self-established phenolic compound database. The analysis process involves the comprehensive application of retention time, mass spectrometry and standard curve of external standard method. The high resolution and precise mass number of HRMS can accurately distinguish the target compounds even if the chromatographic peaks partially overlap. By deeply resolving characteristic ions in the mass spectra, such as molecular ion peaks and fragment ion peaks, combined with retention time information, accurate qualitative analysis of catechins and epicatechin can be achieved. For a more detailed analysis of this part of the data, we described the experimental methods in detail in “Materials and Methods 2.6”. (Line 170-196)

Comments 2: How is the calculation for the phenolics expressed as μg/g for the gastric and intestinal digestion as these media is liquid. is it referred back to the original grain weight?

Response 2: Thank you for your kind comments. The μg/g represents the weight of the various phenolic compounds released per gram of sample, where sample weight refers to the weight of the sample weighed prior to gastrointestinal digestion and colonic fermentation. We have added this information in the manuscript. (Line 194-196)

Comments 3: L 376 what is OTU indices, please defined.

Response 3: Thank you for your kind comments. OTU is an abbreviation for "Operational Taxonomic Unit", a method for classifying microorganisms based on the similarity of their genetic sequences. Typically, 16S rRNA gene sequencing is used to identify different bacterial species and group sequences whose similarity exceeds a defined threshold into the same OTU, representing a putative species or taxonomic unit.

Comments 4: L382 number of OUT … Is this a typo error! it should be OUT! also Check figure 4 as well it is written as OUT.

Response 4: Sorry for the trouble. We have changed the correct writing of OUT.

Comments 5: The manuscript has 6 figures; it may be a good idea to have a supplementary material

Response 5: Thank you for your kind comments. We have added figures related to this study in the supplementary material.

Comments 6: Reference format need to be revised.

Response 6: Thank you for your kind comments. We have revised the reference format.

Reviewer 3 Report

Comments and Suggestions for Authors

The manuscript “Changes of barley bound phenolics and their characteristics during simulated gastrointestinal digestion and colonic fermentation in vitro” is well written, but I have few remarks:

Line 14: could it improve or be toxic to the organism?

Line 22 acetic acid”- again what are the implications? Is good or bad?

The abstract needs more focus on your study importance and implications in physiology

Chapter “In vitro rat simulated gastrointestinal digestion” make clearer that is about the collected fluid, not the rats. I understood the design, but please rephrase it.

Line 187 Do you have the SEM model? Producer? More details about the machine

Table 1 and 2: could you make some correlations between them? Like PCA analysis.

Sem micrographs is there a magnification bar? Seems too small. Same with microscopy images. Maybe add a bar from word/ppt and write with some bigger font the bar dimension

The sequences of all the primers used in pcr? Maybe add a table

Really nice correlation figure, but make it more readable

Really nice manuscript, I really enjoy reading it, hope you will be able to improve it.

Author Response

Comments 1: Line 14: could it improve or be toxic to the organism?

Response 1: Thank you for your kind comments. The phenolic compounds released during gastrointestinal digestion and colonic fermentation exert significant anti-inflammatory, antioxidant, and intestinal barrier-enhancing effects, thereby playing a crucial role in maintaining gastrointestinal health and overall homeostasis. (Line 53-59)

Comments 2: Line 22 acetic acid”- again what are the implications? Is good or bad?

Response 2: Thank you for your kind comments. Acetic acid, a predominant constituent of short-chain fatty acids (SCFAs), is primarily generated through the fermentation of undigested and unabsorbed carbohydrates by anaerobic bacteria in the gut. This metabolite exerts a myriad of physiological functions within the gastrointestinal tract. As a beneficial compound, acetic acid not only serves as an energy source for the host but also actively contributes to intestinal health and overall well-being by modulating the balance of the gut microbiota, lowering intestinal pH, and engaging in various metabolic regulatory processes. (Line 345-350)

Comments 3: The abstract needs more focus on your study importance and implications in physiology.

Response 3: Thank you for your kind suggestions. We have revised parts of the abstract to make the abstract focus more on our study importance and implications.

Comments 4: Chapter “In vitro rat simulated gastrointestinal digestion” make clearer that is about the collected fluid, not the rats. I understood the design, but please rephrase it.

Response 4: We really appreciate your advice. We have redescribed this part of the “In vitro rat simulated gastrointestinal digestion” to make this experimental design easier to understand. (Line 138-152)

Comments 5: Line 187 Do you have the SEM model? Producer? More details about the machine

Response 5: Thank you for your kind comments. The model of the SEM is S-3400N, and the manufacturer is Hitachi, Ltd., Tokyo, Japan, and we have supplemented this information in the manuscript. (Line 199-200)

Comments 6: Table 1 and 2: could you make some correlations between them? Like PCA analysis.

Response 6: Table 1 and Table 2 present the results of quantitative analysis of phenolic compounds released during the gastrointestinal digestion stage and colonic glycolysis stage, respectively. These two tables specifically present the release of phenolic compounds at different stages, so it is not available to directly correlate them by means of principal component analysis (PCA).

Comments 7: Sem micrographs is there a magnification bar? Seems too small. Same with microscopy images. Maybe add a bar from word/ppt and write with some bigger font the bar dimension

Response 7: Thank you for your kind suggestions. We have remodified the images so that the magnification bar can be displayed more clearly.

Comments 8: The sequences of all the primers used in pcr? Maybe add a table

Response 8: Thank you for your kind suggestions. We have confirmed the gut microbiota sequencing method in detail, and the specific method has been modified in Materials and Methods, where pcr primer sequences are included. (Line 224-232)

Round 2

Reviewer 3 Report

Comments and Suggestions for Authors

The authors improved their manuscript quality 'Changes of barley bound phenolics and their characteristics during simulated gastrointestinal digestion and colonic fermentation in vitro ', but they still need to improve the appearence. I know that is hard to have good CLSM images, but maybe you could improve the contrast in those images, and also make the letters in the correlation map bigger you somehow can see the name of the acids, but not of the species.

Author Response

Comment: I know that is hard to have good CLSM images, but maybe you could improve the contrast in those images, and also make the letters in the correlation map bigger you somehow can see the name of the acids, but not of the species.

Response: We sincerely appreciate your positive feedback. We have optimized the image contrast to achieve higher - quality CLSM images. Meanwhile, we have refined the correlation map, enhancing the readability and visibility of its text. (Line 339 and Line 504)